# Adenovirus DNA Polymerase Loses Fidelity on a Stretch of Eleven Homocytidines during Pre-GMP Vaccine Preparation

**DOI:** 10.3390/vaccines10060960

**Published:** 2022-06-16

**Authors:** Zara Hannoun, Edmund G. Wee, Alison Crook, Stefano Colloca, Stefania Di Marco, Tomáš Hanke

**Affiliations:** 1Nuffield Department of Medicine, The Jenner Institute, University of Oxford, Oxford OX3 7DQ, UK; z.hannoun@oxb.com (Z.H.); edmund.wee@ndm.ox.ac.uk (E.G.W.); alison.crook@ndm.ox.ac.uk (A.C.); 2ReiThera S.r.l., Via di Castel Romano, 100, 00128 Rome, Italy; stefano.colloca@reithera.com; 3Advent S.r.l., Via Pontina km 30600, 00071 Pomezia, Italy; s.dimarco@irbm.com; 4Joint Research Center for Human Retrovirus Infection, Kumamoto University, Kumamoto 860-0811, Japan

**Keywords:** adenovirus DNA polymerase, T7 polymerase, polymerase fidelity, protein engineering, vaccines, HIVconsvX

## Abstract

In this study, we invented and construct novel candidate HIV-1 vaccines. Through genetic and protein engineering, we unknowingly constructed an HIV-1-derived transgene with a homopolymeric run of 11 cytidines, which was inserted into an adenovirus vaccine vector. Here, we describe the virus rescue, three rounds of clonal purification and preparation of good manufacturing practise (GMP) starting material assessed for genetic stability in five additional virus passages. Throughout these steps, quality control assays indicated the presence of the transgene in the virus genome, expression of the correct transgene product and immunogenicity in mice. However, DNA sequencing of the transgene revealed additional cytidines inserted into the original 11-cytidine region, and the GMP manufacture had to be aborted. Subsequent analyses indicated that as little as 1/25th of the virus dose used for confirmation of protein expression (10^6^ cells at a multiplicity of infection of 10) and murine immunogenicity (10^8^ infectious units per animal) met the quality acceptance criteria. Similar frameshifts in the expressed proteins were reproduced in a one-reaction *in vitro* transcription/translation employing phage T7 polymerase and *E. coli* ribosomes. Thus, the most likely mechanism for addition of extra cytidines into the ChAdOx1.tHIVconsv6 genome is that the adenovirus DNA polymerase lost its fidelity on a stretch of 11 cytidines, which informs future adenovirus vaccine designs.

## 1. Introduction

Vaccines are the best and most economical solution for the control of infectious diseases [1,2]. Novel virus-vectored vaccine candidates are first constructed in a research laboratory as so-called ‘research-grade’ (RG) vaccines, typically using a synthetic transgene inserted into a safe purpose-engineered virus vector/vaccine platform. These are then characterized for transgene product expression, genetic stability, pre-clinical immunogenicity and, if possible, efficacy in animal models [3,4]. Vaccine candidates passing all quality checks and meeting criteria for further clinical development are transferred for good manufacturing practice (GMP) production in specialized facilities authorized by the national regulator. Only vaccines strictly compliant with the pharmacopeia are allowed to be tested in phase 1 trials in humans [5,6].

Our team is developing an HIV-1 vaccine with the aim of induction of protective killer T cells [7,8,9,10]. Vaccine immunogens (inserts) currently tested in clinical trials are collectively called HIVconsvX and are a family of chimeric proteins assembled from six conserved regions of HIV-1 proteins Gag and Pol [11]. To minimize responses induced by possible junctional neoepitopes irrelevant for HIV protection [12], conserved regions were ordered in six different ways—as 1-2-3-4-5-6 (HIVconsv1), 2-4-6-1-3-5 (HIVconsv2), 3-6-2-5-1-4 (HIVconsv3), 4-1-5-2-6-3 (HIVconsv4), 5-3-1-6-4-2 (HIVconsv5) and 6-5-4-3-2-1 (HIVconsv6)—to avoid boosting with the same junction when used sequentially. Furthermore, we routinely ‘humanize’ the transgene open reading frame (ORF) by replacing the original HIV-1 codons with those used most frequently in human genes, which, for some vectors, such as plasmid DNA, improves the transgene product expression and increases safety and immunogenicity [13,14,15,16].

T cells are best induced by heterologous prime-boost regimens using vectors of two different vaccine modalities [8,17,18,19,20,21,22]. The current priming vector for the HIVconsvX vaccines is ChAdOx1, which is engineered replication-deficient simian adenovirus serotype 23 (Y25) originally isolated from a chimpanzee [23,24]. ChAdOx1 contains a medium-sized, double-stranded DNA genome of approximately 28 kilobase pairs (kbp) [25,26,27]. The mechanisms of adenovirus DNA replication are firmly established, and many of the roles of the key components in the process have been elucidated [28,29,30]. The genome is replicated by adenovirus-encoded DNA polymerase (AdV DNAP) assisted for elongation by viral and cellular proteins [31,32]. A single cell infected with a single virus produces approximately 10^4^ daughter virions [33]. Adenoviruses are important vaccine vectors popular for genetic stability and strong immunogenicity [34,35]. Replication-deficient vectors derived from several adenovirus serotypes have an established manufacture [12,21,36,37,38,39], of which COVID-19 vaccine ChAdOx1 nCoV-19 has been successfully scaled up by AstraZeneca and their partners to a billion-dose production [40].

In the present work, we describe pre-GMP preparation of a candidate HIV-1 vaccine, ChAdOx1.tHIVconsv6 [11]. We were unaware that through codon humanization and region reshuffle, we assembled a continuous run of eleven cytidines in the transgene, which turned out to be the source of genetic instability. The most likely explanation suggested by retrospective analyses was an unexpected loss of fidelity of the AdV DNAP on a stretch of eleven or more cytidines, resulting in extensions of the homocytidine run. Similar frame switching was readily detected in an integrated *in vitro* prokaryotic T7 polymerase transcription/translation system. These observations provide a caution for future passenger transgene engineering for adenovirus-based and probably other vaccine modalities.

## 2. Materials and Methods

### 2.1. Recombinant DNA and Gel Electrophoresis

All enzymes used for PCR and recombinant DNA work were purchased from New England Biolabs (Hitchin, UK) or Boehringer Mannheim (Baden-Württemberg, Germany) and used under the reaction conditions recommended by the vendors. DNA was separated on a 1% agarose and visualized using ethidium bromide and UV light as describe previously [41].

### 2.2. M9 Cells and Rescue of the ChAdOx1.tHIVconsv6 Virus

The history of the adherent GMP-certified M9 cell line derivation is as follows. An ampoule of passage 6 HEK 293 cells frozen at the University of Leiden in 1973 [26] was expanded once and frozen in Frank Graham’s laboratory in 1987 at McMaster University, Hamilton, Canada. This was further expanded to passage 27. In the GMP facility of Advent S.r.l, Pomezia, Italy, a frozen vial of passage 27 was thawed and propagated in culture up to passage 30, when the cells were transfected to stably express tetracycline repressor for the owner, NousCom AG, Basel Switzerland. University of Oxford purchased access to this cell line from NousCom for the GMP manufacture of the ChAdOx1.tHIVconsv6 vaccine. M9 cells were maintained in DMEM10 medium (DMEM medium supplemented with L-glutamine and 10% foetal bovine serum (FBS) (GIBCO; #16250) in a humidified incubator at 37 °C, 5% CO_2_. Adherent M9 cells were adapted to growth in a cell suspension in chemically defined medium CD293 (Invitrogen) by a gradual reduction of FBS and designated M9.S.

To rescue the recombinant ChAdOx1.tHIVconsv6 virus, linear genomic DNA was excised from bacterial artificial chromosome (BAC) p4056 [11] using the PmeI restriction endonuclease, and 5 µg was transfected into the adherent M9 cells using Lipofectamine^TM^ CD2000 transfection reagent (ThermoFisher Scientific, Pittsburgh, PA, USA) in a 25 cm^2^ tissue flask (T25) following the vendor’s instructions as described previously [11]. The virus was harvested 6 days later, when a cytopathic effect was observed, by collecting all the cells and cell culture supernatant. These were frozen and thawed three times and used for reinfection of 175 cm^2^ of a fresh 80% confluent cell monolayer (Figure 1). The ChAdOx1.tHIVconsv6 RG virus was first prepared by the Vector Core Facility of the Jenner Institute, University of Oxford, and characterized prior to transfer of the BAC p4056 DNA to Advent S.r.l, Pomezia, Italy, where all the work on the ChAdOx1.tHIVconsv6 pre-GMP material described here was carried out.

### 2.3. Single Virion-Infection (SVI) Purifications

For the ChAdOx1.tHIVconsv6 virus rescue and serial SVI clonal purification, M9 cells were grown in the DMEM10 medium in a humidified incubator at 37 °C, 5% CO_2_. Briefly, passage-1 virus was diluted and dispensed onto an 80% confluent M9 cell monolayer in 96-well plates (96wp) with an average of 3 and 1 infectious units (IU) seeded per well. Over the following couple of days, cell monolayers were visually inspected for virus plaques, and only wells with a single plaque were harvested by scraping the cell monolayer and freezing both the cells and the supernatant together. The whole 96wp content was then expanded in a 24wp, followed by a T75 flask, tested for the concentration of virus particles (vp) and IU and assessed for expression of the transgene product of the correct relative molecular mass (*M_r_*) using a Western blot. For methods used for determining the virus concentration and for assessment of expression, please refer to the corresponding methods sections. At the end of the SVI process, three clones were selected, diluted and used for the next round (see Figure 1).

### 2.4. Identity, Flank-to-Flank and Purity PCRs

The workflow consisted of isolation of viral DNA from 200 µL of the test virus stock (QIAamp DNA blood mini kit, Cat. 51106, Qiagen, Milano, Italy), assembly and running of a PCR reaction using extracted viral DNA as the template (Eppendorf GS Master Cycle, Milano, Italy), analysis by electrophoresis on agarose gel, image capture using trans-illuminator and verification of the sample pattern.

Identity (ID) PCR uses a combination of out-of-the-insert (or flank) and transgene-specific primers and confirms the presence of a transgene-specific primer complementary sequence within the transgene and its distance to the outer primer-binding site in the virus but does not confirm the presence of the full-size transgene. The expected size of the ID PCR product for ChAdOx1.tHIVconsv6 was 1738 bp. Primers: ChAdOxLeftEnd GCTCATGATGCTGCAACACAC, ChAdOxRev3-CATGGGCCTGAACAAGATCG; cycling conditions: primary denaturation at 95 °C for 3 min, (denaturation at 95 °C for 40 s, annealing at 60 °C for 15 s, extension at 68 °C for 2 min) 25×, final extension at 72 °C for 3 min, end.

Flank-to-flank PCR uses primers binding up- and downstream of the whole DNA insert carrying the transgene ORF and serves as a screen for the presence of the full-size transgene. The expected size of the flank-to-flank PCR product for ChAdOx1.tHIVconsv6 was 3172 bp. Primers: SubmonteFor GCTCATGATGCTGCAACACAC, VJdown-GCTCATGATGCTGCAACACAC; cycling conditions: primary denaturation at 95 °C for 3 min (denaturation at 95 °C 40 s, annealing at 60 °C for 15 s, extension at 68 °C for 3 min) 25×, final extension at 72 °C for 3 min, end.

Purity PCR confirms the absence of non-viral plasmid sequences (NVPSs) in the virus preparation. NVPSs ensure BAC stability and replication in the host bacterium and supply a chloramphenicol resistance gene marker for selection of transformed bacteria. Whereas inclusion of NVPS should not allow adenovirus replication because of the requirement for the adenovirus terminal repeats at each end of the double-stranded DNA genome, NVPSs are occasionally detected in virus preparations. Here, clones positive for NVPS PCR were discarded. The expected size of the NVPS PCR product was 759 bp. Primers: CMFor CGGGCGTATTTTTTGAGTTATCG, CMRev-CAGGCGTAGCAACCAGGCG; cycling conditions: primary denaturation at 95 °C for 3 min (denaturation at 95 °C for 40 s, annealing at 60 °C for 15 s, extension at 68 °C for 1 min) 25×, final extension at 72 °C for 3 min, end.

### 2.5. SDS-PAGE and Western Blotting

HeLa cells were infected with ChAdOx1.tHIVconsv6 at multiplicity of infection (MOI) of 10 and lysed 48 h later. Briefly, cells were placed on ice, and cold lysis buffer (20 mM Tris pH 8.0, 137 mM NaCl, 10% glycerol, 1% NP40) was added. The cells were then scraped, transferred to a 1.5 mL Eppendorf tube, vortexed, incubated on ice for 1 h and spun at 13,000× *g* at 4 °C for 10 min. Soluble proteins were separated using two identical SDS-polyacrylamide gels cross-linked with 15% (*N*,*N*-diallyltartardiamide) using a Novex Nu-PAGE system (ThermoFisher Scientific, Pittsburgh, PA, USA) and either transferred onto a nitrocellulose filter (Amersham International, Stafford, UK) utilizing a Transfer Blot Turbo system (BioRad, Hercules, CA, USA) or stained with Coomassie blue. Membrane-bound proteins were incubated with primary anti-HIV-1 Gag p24 mAb ab9071 (Abcam, Cambridge, UK), followed by anti-mouse IgG secondary antibody conjugated to horse radish peroxidase (HRP) (715-035-150, JacksonImmunoResearch Laboratories, West Grove, PA, USA) and enhanced chemiluminescence (ECL; Amersham International, Stafford, UK). Furthermore, human antibody 91-5 [42] was obtain through the NIH AIDS Reagent Program and used as an alternative primary reagent, followed by anti-human HRP-conjugated mAb and ECL. *In vitro* transcription/translation reactions (see Section 2.11) were loaded directly on the gel and processed as above. RG ChAdOx1.tHIVconsv6 virus-infected HeLa cell lysate (MOI 100) was used as a positive control.

### 2.6. Virus Preparation

Following each round of SVI clonal purification and expansion in 12 T25 flasks (Figure 1), viruses were released from harvested cells by detergent (1% TWEEN-20) lysis and clarified by depth filtration. AEX chromatography was used for the virus seed stock (VSS) purification. The titre was determined by real-time quantitative PCR (QPCR) employing TaqMan reporter–quencher dye chemistry using a probe labelled with a reporter dye (6-FAM) at the 5′ end and a quencher dye (TAMRA) at the 3′ end and an ABI Prism 7900HT sequence detection system for the vp concentration. An infectivity assay was performed with an anti-hexon antibody/anti-mouse mAb conjugated to horse radish peroxidase and the 3, 3′-diaminobenzidine (DAB) substrate, resulting in a dark brown product to stain the infected cell monolayers and determine the IU concentration. All procedures were performed strictly according to the Advent manufacturer’s proprietary SOPs.

### 2.7. DNA Sequencing of the Transgene

The sequence of the whole transgene region in the VSS was initially assessed by direct Sanger sequencing of the phenol/chloroform purified viral genomic DNA [41]. Then, using primers For1637 TCGAGGAGCTGCGCCAG, Rev2424 TCAGGCCATGATGATC and Rev2109 GTGGCGCCCTCGCTCAG, the presence of 12 cytidines was verified with a hint of a mixed template. For clonal analysis of the VSS, the VSS genomic DNA was phenol/chloroform-extracted and digested with BamHI and SalI, and the fragment containing the 11-C run was ligated into pUC19. Correct clones were identified by NotI and BamHI diagnostic digests and sequenced with primers EW101FOR: GCCCGAGAAGGACAGCTGGACCG (250 bp upstream of 11-C region) and EW101REV: CCTTCAGCATCTGCATGGCGGC (250 bp downstream of 1-C region) [41].

### 2.8. Animals, Vaccinations and Preparation of Splenocytes

Six-week-old female BALB/cJ mice (Charles River, Harlow, UK) were immunized intramuscularly under general anaesthesia with a total 10^8^ IU of ChAdOx1.tHIVconsv6 preparations, and the T-cell responses were analysed 9 days later. On the day of sacrifice, spleens were collected, and cells were isolated by pressing organs individually through a 70 µm nylon mesh sterile cell strainer (Thermo Fisher Scientific, Waltham, MA, USA) using a 5 mL syringe rubber plunger. Following the removal of red blood cells with RBC Hybri-Max lysing buffer (Sigma-Aldrich Company, Pool, UK), splenocytes were washed and resuspended in R10 (RPMI 1640 supplemented with 10% FCS, penicillin/streptomycin and β-mercaptoethanol) for the ELISPOT assay.

### 2.9. Peptides

All peptides used in the ELISPOT assay were >90% pure according to mass spectrometry (Synpeptide, Shanghai, China), were dissolved in DMSO (Sigma-Aldrich, Pool, UK) to yield a stock of 10 mg/mL and stored at −80 °C until use.

### 2.10. IFN-γ ELISPOT Assay

An enzyme-linked immunospot (ELISPOT) assay was performed using a mouse IFN-γ ELISpot kit (Mabtech, Stockholm, Sweden) according to the manufacturer’s instructions as described previously [43]. Immune splenocytes were collected and tested separately from individual mice. Peptides were used at a concentration of 2 μg/mL each, and splenocytes at a concentration of 10^5^ cells/well were added to 96wp high-protein binding Immobilon-P membrane plates (Merck Millipore, Gillingham, UK) that had been pre-coated with 5 µg/mL anti-IFN-γ mAb AN18 (Mabtech). The plates were incubated at 37 °C in 5% CO_2_ for 18 h and washed with phosphate-buffered saline (PBS) before the addition of 1 µg/mL biotinylated anti-IFN-γ Mab (Mabtech) at room temperature for 2 h. The plates were then washed with PBS, incubated with 1 µg/mL streptavidin-conjugated alkaline phosphatase (Mabtech) at room temperature for 1 h and washed with PBS. Individual cytokine-producing units were detected as dark spots after a 10-min reaction with 5-bromo-4-chloro-3-idolyl phosphate and nitro blue tetrazolium using an alkaline phosphatase-conjugated substrate (Bio-Rad, Richmond, CA, USA). Spot-forming units were counted using an AID ELISpot reader system (AID Autoimmun Diagnostika, Baden-Württemberg, Germany). The frequencies of responding cells were expressed as number of spot-forming units (SFU)/10^6^ splenocytes.

### 2.11. Cell-Free Transcription/Translation

Test tube transcription/translation experiments were conducted using a PURExpress^®^
*in vitro* protein synthesis kit (New England Biolabs, Hitchin, UK; Cat. No NEB #E6800) according to the vendor’s protocol.

## 3. Results

### 3.1. Construction of the BAC Carrying the ChAdOx1.tHIVconsv6 Vaccine Genome

The tHIVconsv6 ORF coded for 895 amino acids of the immunogen. The original HIV-1 ribonucleic acid sequence was ‘rewritten’ into DNA using the most frequent human codons, and its six subregions were arranged in the 6-5-4-3-2-1 succession [11]. A 2696 bp DNA fragment carrying the tHIVconsv6 ORF was inserted into the E1 locus of the ChAdOx1 genome contained in BAC. The correct DNA sequence was confirmed for the entire inserted DNA fragment in the BAC.

### 3.2. Rescue, Clonal Purification and Preparation of the Virus Seed Stock of the Vaccine Virus

The workflow, starting from the virus rescue, initial amplification, three rounds of single-virion infection, i.e., clonal purification, and the final expansion to generate the VSS, which constituted the starting material for the GMP manufacture, is depicted in Figure 1. The passage-1 virus expanded sufficiently to determine the titre and confirm the tHIVconsv6 protein expression in infected HeLa cells using mAb specific for Gag p24 present in C-terminal region 1 (Figure 2a). HeLa cells were used as a non-permissive human cell line for ChAdOx1.tHIVconsv6 replication and therefore like the cells in vaccine recipients producing the immunogen. Three rounds of clonal purification followed, and all expanded clones were tested for protein expression and transgene presence by ID and flank-to-flank PCRs (only one of two is shown in Figure 2b–d for each step for illustration) after each purification. The absence of non-viral plasmid sequence was confirmed by a lack of PCR product in the NVPS PCR (not shown). Following SVI-3, clone 3BG1.1AC11.1AE1 was selected for preparation of the VSS (Figure 2e), which required a further two passages. In total, the VSS went through 12 virus passages from the DNA transfection to the VSS.

### 3.3. ChAdOx1.tHIVconsv6 VSS Passed All but the DNA Sequence Quality Tests

To test the vaccine genetic stability, the VSS was passaged five times in suspension M9.S cells used for the drug substance preparation. The generated virus stock VSS+5 readily expressed the transgene product upon infection of HeLa cells (Figure 3a). In parallel, the final passage of the VSS necessary for production of the clinical batch material was optimized for the M9.S cell density at infection, multiplicity of infection and time of harvest to maximize the virus yield. These parameters were confirmed in the engineering run, which mimics the GMP run at the same scale, generating material suitable for formal pre-clinical toxicity testing in animals (ToxLot) (Figure 3b). Vaccination of BALB/cJ mice with the VSS, VSS+5, ToxLot and the laboratory-produced RG ChAdOx1.tHIVconsv6 induced comparable CD8^+^ T-cell frequencies, recognizing eight well-defined murine epitopes in an IFN-γ ELISPOT assay, including epitope AMQMLKETI in Gag p24 [44] (Figure 3c,f). However, DNA sequencing of the VSS ORF detected additional cytidine in a stretch of eleven cytidines and aborted the clean room manufacture (Figure 3d). Still, two different mouse ab9071 and human 95-1 monoclonal antibodies specific for HIV-1 Gag p24 readily detected region 1 (Gag p24) downstream of the 11-cytidine junction and visualized the correct full-size tHIVconsv6 protein (Figure 3e,f) in the VSS-infected cell lysates. To eliminate the possibility that the additional cytidine was a technical problem arising from direct sequencing of the adenovirus genome of the VSS, genomes of other virus clones after SVI-3 were sequenced. All confirmed the presence of 12 cytidines (Figure 3d) using multiple primers. Therefore, we concluded that AdV DNAP loses fidelity on a stretch of 11 cytidines—the most likely explanation.

### 3.4. The VSS Is a Mixed Population

To explain expression of the full-length tHIVconsv6 and immunogenicity of the p24 AMQ epitope in mice, a DNA fragment with the region 2–region 1 junction was excised from the VSS genome and subcloned into a plasmid for sequencing. The first 20 bacterial colonies yielded 16 clones with 12 cytidines and 4 clones with 13 cytidines but no clone which would allow translation of the correct p24. Sequencing of 100 colonies revealed clones with 11, 12, 13, 14 and 15 cytidines in 1, 70, 25, 3 and 1 instances, respectively (Figure 3g), allowing transgene products with Gag p24 to be made from clones with 11 and 14 cytidines. No clone with fewer than 11 cytidines was detected. This demonstrated that the observed protein production and induction of T cells *in vivo* was provided by 1/25^th^ (4 correct clones out of 100) of the tested ChAdOx1.tHIVconsv6 virus preparation. Schema of the ChAdOx1.tHIVconsv6 vaccine developmental path from the plasmid DNA to the preparation of the VSS is shown in Figure 4.

### 3.5. In Vitro Transcription/Translation System Slips Regularly on Long Polycytidines

To demonstrate slippage on long runs of cytidines, we employed a PURExpress^®^ cell-free transcription/translation system. Three plasmids were constructed using the kit-provided control plasmid carrying an ORF coding for the *E. coli* dihydrofolate reductase (DFHR), to which a run of 11, 12 or 13 cytidines was coupled, followed by the Pk (also known as SV5), Myc and HA (influenza virus A hemagglutinin) tags recognized by monoclonal antibodies in the forward three reading frames, thus monitoring a frame switch during transcription/translation. Using Western blot, tag-specific mAbs-visualized proteins in all three reading frames translated from all three input plasmids in similar amounts, indicating that bacterial T7 RNA polymerase also lost its bearings on long cytidine stretches and/or the *E. coli* ribosome slipped on the mRNA template with regularity (Figure 5).

## 4. Discussion

We designed novel transgenes for candidate subunit vaccines. Through a combination of gene and protein engineering, we introduced a continuous run of 11 cytidines (joining 8 cytidines from the 3′-end of region 2 with 3 cytidines from the 5′ end of region 1) into the immunogen tHIVconsv6 (6-5-4-3-2-1) ORF [11]. This homocytidine sequence resulted in a rare but regular addition of cytidines, most likely by the AdV DNA polymerase, and the ChAdOx1.tHIVconsv6 vaccine manufacture had to be aborted. The region 2–region 1 junction was the only 11-cytidine sequence in the tHIVconsv6 transgene that had no stretches of 10, 9 or 8 cytidines and contained two runs of 7 genetically stable cytidines. It is of note that vaccines ChAdOx1.tHIVconsv1 (1-2-3-4-5-6) and poxvirus MVA.tHIVconsv4 (4-1-5-2-6-3) contain one stretch of nine cytidines each, showed no loss of the AdV and poxvirus polymerase fidelity and have both been successfully manufactured (unpublished T.H., E.G.W, A.C. and S.D.M.). To rectify the tHIVconsv6 gene instability, an ORF coding for immunologically equivalent (the same T-cell epitopes) version 2 of the immunogen, called HIVconsv62 (6-5-4-3-2-1/1), was synthesized with many CCC proline codons altered to CCA or CCT to avoid any stretches of more than four cytidines [43]. Vaccine ChAdOx1.HIVconsv62 is stable and has now been successfully manufactured (unpublished T.H., E.G.W., A.C. and S.D.M.). BAC, a single-copy plasmid with a low level of replication in *E. coli*, contained the entire ChAdOx1.tHIVconsv6 genome with the 11 homocytidine run and appeared to be genetically stable, at least on the plasmid miniprep scale employed in the present work.

We have reproduced similar frameshifting in a one-step reaction in a highly controlled PURExpress^®^ cell-free system reconstituted from a single-subunit RNA polymerase of the T7 bacteriophage (T7 ssRNAP) and purified components necessary for the *E. coli* translation [45]. In this system, both DNA-programmed RNA polymerase and ribosome slippage could have contributed to the frameshifting. As for the polymerase error, polymerase slippage was historically first demonstrated in *E. coli* [46]. The crystal structure of the elongation complex revealed that T7 ssRNAP holds a DNA-RNA hybrid of 7-8 bp [47], which would be compatible with losing a reference on an 11-homocytidine run. Thus, a similar loss of reference resulting in a longer copied strand can happen to both DNA-dependent DNA and RNA polymerases, whereby their proofreading function normally efficiently correcting single-stranded 3′ end mismatched nucleotides cannot resolve this, as no mismatch results from the slippage. Repetitive addition of nucleotides to the 3′ end of nascent transcripts due to an upstream backward slippage is called reiterative transcription or polymerase stuttering and is one of many types of transcription control mechanisms of bacterial, viral and eukaryotic polymerases [48]. It can also be involved in human disease [49]. Note that adenoviruses use a backward slippage during the terminal protein priming of their genomic DNA replication [28]. Interestingly, when the effect of the template length, primer length and (homo)template composition on the fidelity of the avian myeloblastosis virus reverse transcriptase was studied on artificial templates, Falvey and colleagues reported that the transcription of oligoribocytidine was the only one of the ribonucleotides tested that was faithful with respect to the DNA product length, whereas slippages occurred on oligo(rA) and oligo(rI), resulting in considerably longer DNA products than the template; oligo(rU) stalled the elongation reaction completely [50].

The second mechanism that could contribute to frameshifting in the PURExpress^TM^ reaction is translational frameshift of the ribosome on the mRNA template. For example, this is used by several RNA viruses to express their polymerase genes and usually consists of two components: a shifty sequence that facilitates ribosome slippage and a signal, either a secondary mRNA structure or unfilled ribosome site A, which stalls the ribosome and increases the likelihood of slippage [51].

## 5. Conclusions

In conclusion, both polymerase and ribosome slippage could have contributed to the correct tHIVconsv6 expression and the frameshifting in the PURExpress^TM^ reaction. Although we have not formally proven that it was the AdV DNAP that made the errors, additional cytidines incorporated into the ChAdOx1.tHIVconsv6 genome point very strongly to this enzyme as the culprit. We find other explanations for additional Cs, such as clonal expansion from mutated BAC, highly unlikely because the VSS is a mixed population of at least five species of homocytidine stretches (11 to 15), yet its preparation included three cycles of single-virion infection, i.e., clonal purification. This strongly supports the argument against a mixed population of the initial transfected plasmids and their clonal propagation. In fact, we put the pre-GMP vaccine through three SVIs precisely to avoid a mixed population. Thus, we conclude that the AdV DNAP lost its fidelity on stretches of 11 homocytidines and incorporated additional cytidines into the copied nucleic acid strand with a low but likely consistent rate. Now that adenovirus-based vaccines have reached the market and proved the utility of this platform for infectious diseases, it is relevant for vaccine developers to be aware of this problem. Long homonucleotide sequences are easily avoidable in synthetic ORFs by using alternative codons for lysine (AAA), glycine (GGG), proline (CCC) and phenylalanine (TTT).

## Figures and Tables

**Figure 1 vaccines-10-00960-f001:**
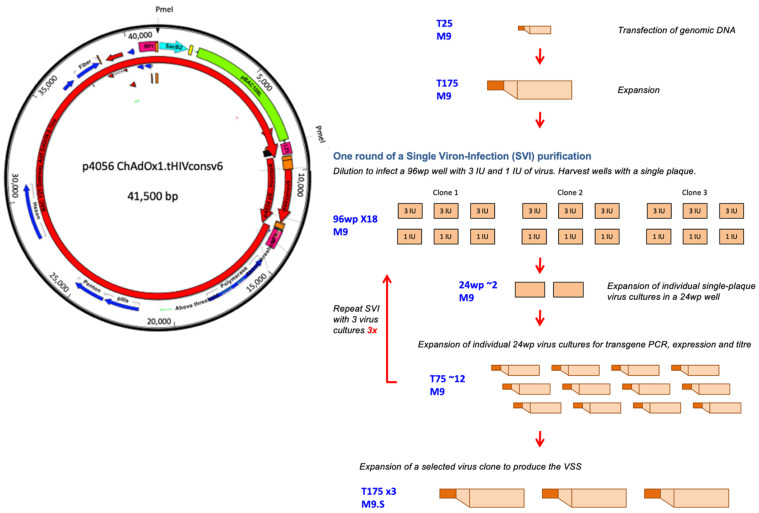
Preparation of the virus seed stock. (**Left**) Schematic map of BAC p4056. The dsDNA ChAdOx1.tHIVconsv6 genome was excised from BAC p4056 using the PmeI restriction endonuclease. (**Right**) The VSS preparation started transfection of the genome into M9 cells in T25, and the whole cell culture harvest was amplified in T175. The resulting virus preparation was titred and tested by ID, flank-to-flank and purity PCRs, as well as for transgene product expression using Western blot of infected cell lysates assessed for Gag p24-specific mAb reactivity and size. The first round of SVI purification started by infection of 96wp cultures with an average of either 3 or 1 IU per well. The cell monolayers were inspected regularly over the following week, and only wells with a single visible plaque were collected and expanded first in 24wp wells and then in T75 to make enough virus for titration, PCR and expression testing. Three virus clones were taken into the next round of SVI. After SVI-3, the best growing/expressing clone was expanded in three T175 flasks of M9.S cells to generate the VSS, which served as the GMP starting material.

**Figure 2 vaccines-10-00960-f002:**
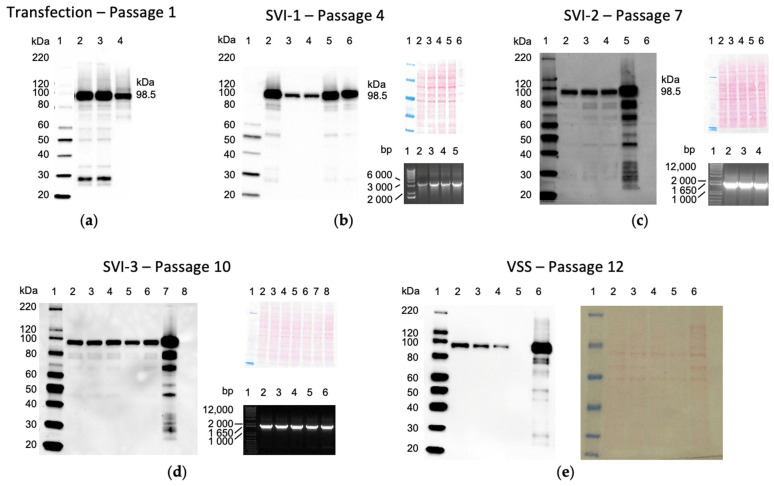
Control of critical steps and process intermediates. At critical stages of the clonal purification and expansion cycles, virus cultures were tested for tHIVconsv6 protein expression in infected HeLa cells using anti-HIV-1 Gag p24 mAbs (ab9071) specific for the C-terminal region 1 in a Western blot (**a**–**e**). The amount of loaded protein in each lane was visualized by direct protein staining (**b**–**e**). Transgene presence was assessed by flank-to-flank (**b**) and ID (**c**,**d**) PCRs (only one or the other is shown). (**a**) Passage-1 cells were transfected with BAC-excised linear ChAdOx1.tHIVconsv6 genomic DNA, the rescued virus was expanded by one passage and the transgene expression was confirmed. *M_r_* markers in kDa (lane 1); MOI 500 (lane 2); MOI 1000 (lane 3); and RG vaccine ChAdOx1.tHIVconsv6 was used as a positive control (lane 4). (**b**) Passage 4. Following SVI-1, tHIVconsv6 expression (left) and transgene presence (bottom right) were confirmed for clones 1AC8 (lane 2), 1BE4 (lane 3), 3BA5 (lane 4), 3BG1 (lane 5) and RG (lane 6). (**c**) Passage 7. SVI-2 yielded clones 1BE4.3AG3 (lane 2), 3BG1.1AC11 (lane 3) and 3BG1.1BE5 (lane 4), which were compared to the RG vaccine (lane 5) and uninfected HeLa cell lysate (lane 6) for HIVconsv6 expression (left) and transgene presence by ID PCR (bottom right). (**d**) Passage 10. Following SVI-3, the transgene product expression (left) and transgene presence (bottom right) were analysed for clones 3BG1.1A11.1AE1 (lane 2), 1BE4.3AG3.1CC4 (lane 3), 3BG1.1BE5.3AA5 (lane 4) and 1BE4.3AG3.1AD12 (lane 5). SVI-2 clone 3BG1.1BE5 (lane 6), the RG vaccine (lane 7) and uninfected Hela cells (lane 8) were used as controls. (**e**) Passage 12. The VSS was generated from 3BG1.1AC11.1AE1, and tHIVconsv6 expression (lane 2) was compared to SVI-3 clone 3BG1.1AC11.1AE1 (lane 3), SVI-2 clone 3BG1.1AC11 (lane 4), uninfected Hela cells (lane 5) and the RG vaccine (lane 6) (left), with the total loaded protein visualized (right).

**Figure 3 vaccines-10-00960-f003:**
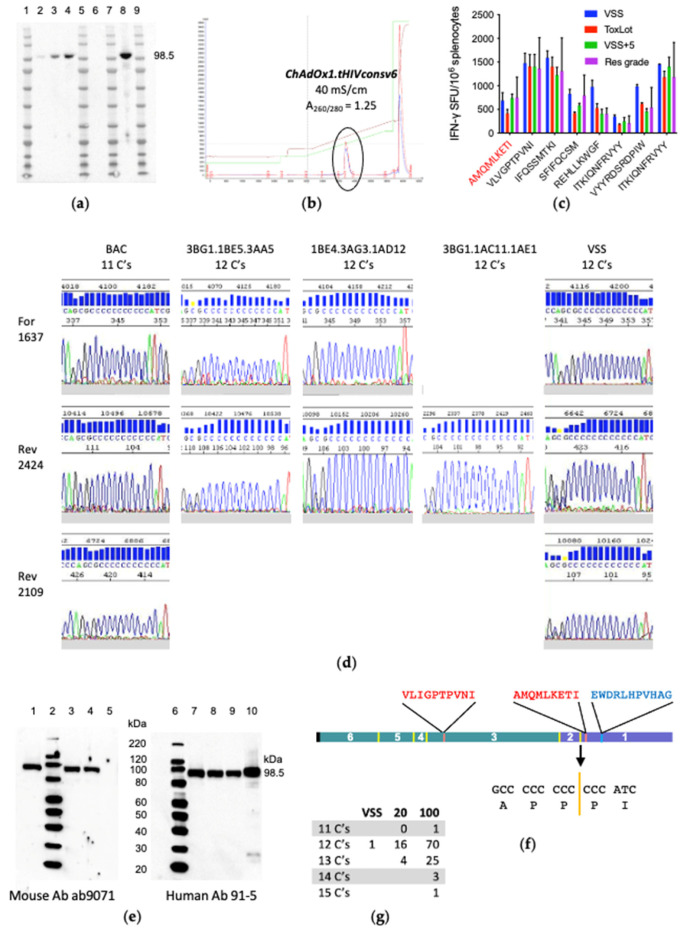
The VSS quality control. (**a**) The VSS established from clone 3BG1.1AC11.1AE1 was tested for genetic stability over five blind passages in the M9.S suspension cells (VSS+5), and the tHIVconsv6 protein expression was assessed in HeLa cells infected at vp MOIs of 2, 20 and 200 (lanes 2–4). Protein expression was compared to uninfected (lane 6) and RG vaccine-infected (lane 8) cells in a Western blot using HIV-1 Gag p24-specific mAb ab9071. (**b**) Using one cell passage from the VSS, material for the pre-clinical toxicity study in an animal model (ToxLot) was prepared and purified using AEX chromatography. A single virus peak was eluted at wavelengths of 260 nm (red) and 280 nm (blue). (**c**) Mouse immunogenicity. The VSS, ToxLot, VSS+5 and the RG vaccine were used to immunize BALB/cJ mice intramuscularly at a dose of 10^8^ IU per animal, and the elicited T cells were tested for recognition of eight well-defined epitopes in tHIVconsv6, of which the AMQMLKETI is downstream of the mutated polycytidine region. Reactive splenocytes were enumerated as SFU in an IFN-γ ELISPOT assay. Mean ± SD (*n* = 3) are shown. (**d**) DNA sequencing traces across the 11 cytidine regions of the starting BAC DNA, three SVI-3 clones and the VSS using one forward and two reverse primers indicated on the left. (**e**) tHIVconsv6 detection in Western blot by two HIV-1 Gag p24-specific antibodies given below in HeLa cells infected with the VSS (lanes 1 and 7), SVI-3 clone 3BG1.1AC11.1AE1 (lanes 3 and 8), SVI-2 clone 3BG1.1AC11 (lanes 4 and 9), uninfected (lane 5) and the RG vaccine (lane 10). (**f**) Schematic representation of the tHIVconsv6 protein indicating its regions, 6-5-4-3-2-1, originating from Pol (green) and Gag (navy blue); the regional junctions (yellow), including junction 2-1 generating 11 cytidines and two CD8^+^ T cells (red); and one mAb 91-5 (blue) epitopes used in the analyses. (**g**) Initially, the entire transgene in the VSS was sequenced directly from the purified genome (VSS). Then, the 11-C fragment was excised from the VSS genome and subcloned into pUC19. Next, 20 and, later, 100 bacterial colonies were sequenced. The table indicates the lengths of the cytidine run and their frequencies.

**Figure 4 vaccines-10-00960-f004:**
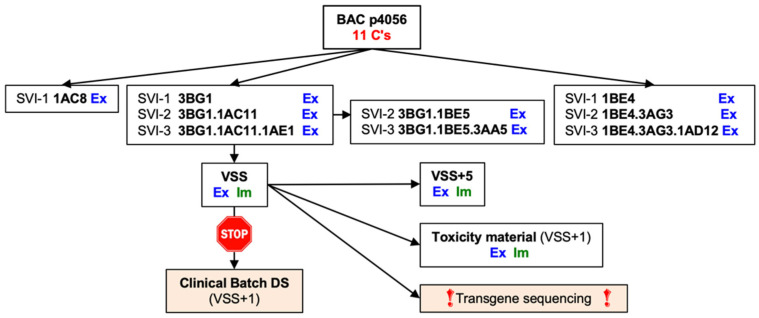
Workflow of the aborted pre-clinical development of the ChAdOx1.tHIVconsv6 vaccine. Numbers in the boxes indicate the clone numbers after SVI-1 (X), SVI-2 (X.X) and SVI-3 (X.X.X). BAC—bacterial artificial chromosome; SVI—single virion infection; Ex—expression of tHIVconsv6 protein with p24 confirmed in infected HeLa cells; Im—murine immunogenicity to 8 defined epitopes, of which AMQMLKETI is in Gag p24 downstream of the junctional 11-cytidine run; VSS—virus seed stock; VSS+1—VSS passaged one time in the M9.S cells; VSS+5—VSS plus five blind passages in the M9.S cells; DS—drug substance is the crude virus harvest at the end of the upstream process.

**Figure 5 vaccines-10-00960-f005:**
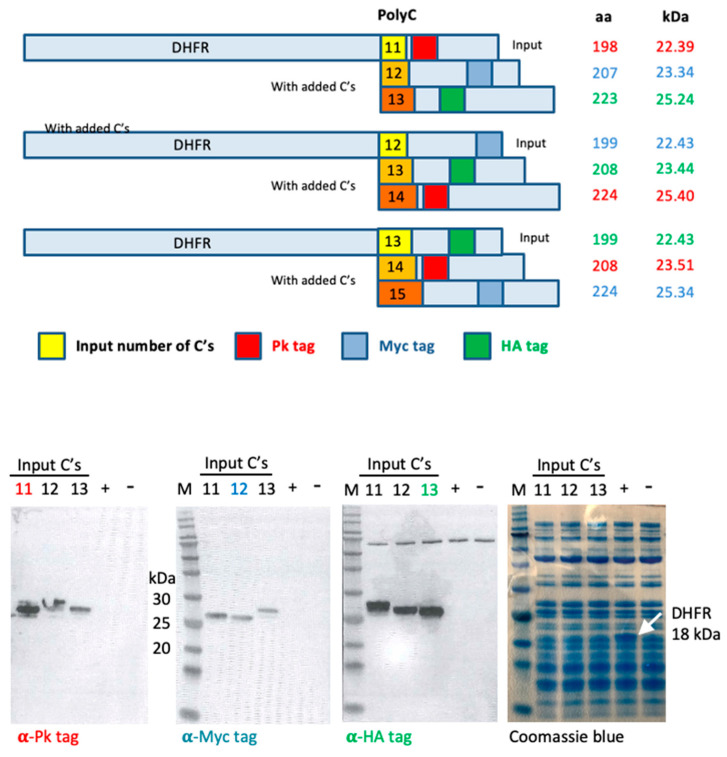
Slippage during a one-step *in vitro* transcription/translation. (Top) A schematic representation of DHFR ORF coupled to 11, 12 or 13 homocytidine runs and mAb tags designated Pk (aka SV5), Myc and HA (influenza virus A hemagglutinin) in three reading frames. The amino acid lengths of the expressed proteins, as well as their estimated *M_r_* in kDa, are shown next to the schematics. The ORF nucleotide and predicted protein amino acid sequences are available in Appendix A. (Bottom) An *in vitro* T7 polymerase transcription/*E. coli* translation reaction was conducted with each of the plasmid ORFs indicated above as the template, followed by a Western blot analysis of the three separately run reactions loaded next to each other. Three identical Western blot membranes were incubated, with individual tag-specific mAbs indicated below, and a secondary enzyme-conjugated mAb, followed by ECL. The amount of loaded protein is given in the right-most panel. The predicted *M_r_* of DHFR was 18 kDa.

## Data Availability

All data are contained within the article.

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
