# Peer review of "Adenovirus DNA Polymerase Loses Fidelity on a Stretch of Eleven Homocytidines during Pre-GMP Vaccine Preparation"

_vaccines, 2022, doi:10.3390/vaccines10060960_

Round 1

Reviewer 1 Report

In this manuscript, the authors reported loss of the fidelity of adenovirus DNA polymerase and the underlying mechanisms. The gene instability was observed if there exist long homonucleotide sequences, especially a stretch of eleven homocytidines, in the transgene inserted into adenovirus vector. The resulting frameshifting mutations produced as little as 1/25th of the virus stock contributing to the protein expression and murine immunogenicity in the quality monitoring test, which was recaptured in a one-reaction in vitro prokaryotic T7 polymerase transcription-translation system. Hence, the upstream backward slippage of adenovirus DNA polymerase during the terminal protein priming of their genomic DNA replication is the likely explanation for the framshifting mutations. Considering the increasing development of adenovirus-vectored vaccines, this work is of great significance for the design and application of qualified adenovirus vaccine. But, some problems still need to be solved before it is accepted for press.  

The major concerns:

  1. Page 5, figure 2e, as the introduction in 2.6 from the section of Materials and Methods, it seems that HeLa cells were infected by the same 10 MOI of each recombinant ChAdOx1 from different stages, why the expression levels are so different that the decreased amount observed from VSS, 3BG1.1AC11.1AE1 to 3BG1.1AC11?
  2. Page 6, lines 230 – 234, why do you say AMQMLKETI in Gag p24, which is different from the displayed position in figure 3f. Upon the colour, isn’t it located in the fragment colored with navy blue meaning the pol peptide?
  3. Page 6, lines 235 – 240, it seems better to remove this paragraph above and immediately follow the introduction about the data of figure 3d. Additionally, the description “various stages of SVI” seems contradictory with the data in the figure 3d, all the three sequenced genomes of other virus clones all came from the same stage of the third round.
  4. Page 7, lines 253 – 255, in the section of materials and methods, it should introduce the method about the DNA sequencing of the genomes containing 11 cytidine regions, such as the preparation of the sequenced genome samples. Why are the positions of the 11 or 12 cytidine regions indicated by the No of DNA sequence so different, considering the same primer used and the assayed sequence located just in one sequencing reaction?
  5. Pages 7 – 8, lines 262 – 264, clearly, this sentence wasn’t written or placed suitably.
  6. Page 8, line 282, please see the No2 question under this concerns.
  7. Page 9, line 298 – 300, it is very much confusing on the current expressions of +1 and +2. It was suggested to add the colour symbol indicating the DHFR or the modified DHFR, depicture the more detailed information of the frameshifting and the expression of the individual tags starting from 11, 12, or 13 homocytidine runs.

The minor concerns:

  1. Page 7, line 256, lanes 1 and 6 should be corrected as 1 and 7.
  2. Page 7, line 262, was the polycytidine region excised by the restriction endonuclease? What are them?
  3. Page 8, line 274, it was suggested to provide the calculation method or the introduction for this evaluation of 1/25th.
  4. Page 8, line 283, is or in?
  5. Page 10, line 366, please delete “Lesson learned”.

Reviewer 2 Report

Review of the manuscript entitled “Adenovirus DNA polymerase loses fidelity on a stretch of eleven homocytidines” (vaccines-1673713).

            In this work, the description of the pre- Good Manufacturing Practice (GMP) production of a potential vaccine against HIV-1 based on a non-replicative chimpanzee adenoviral vector expressing a HIV-1 transgene previously obtained, serves to detect the error in the transcription/traduction of DNA frames with continuous run of 11 or more cytidines. Adenoviral polymerase seems to add regularly more cytidines in that frames. GMP process is really important to finally have a vaccine on the market.

            The work describes well the necessary steps to have a ready-to-use vaccine stock, but that use is paralyzed with the chosen candidate. Regarding the ability of the adenovirus polymerase to erroneously introduce more C in a fragment greater than 9C, it is a somewhat scarce contribution, although I consider it useful. In my opinion, some in vitro assay with the purified polymerase would be necessary in order to really attribute this failure to it and give weight to the work presented. The authors themselves do not quite specify whether the error is due to the polymerase or the translation in the ribosomes.

The work is well include in this special issue. Some aspects can be improved by answering the questions detailed below:

  • Why have the authors selected this ChAdOx1-tHIV consv6 vaccine candidate? Have also the other transgenes mentioned in the Introduction regions with multiple cytidines? Thus, as mentioned in the Discussion (lines 316-318), do the authors consider the problem for the Ad Polymerase only in regions with more than 9 cytidines?

  • Please, add in Materials and Methods the name, origin, media for growth etc... of cells used for the SVI assay.

  • Could you explain the selection of HeLa cells for quality control assays?

  • How the authors explain so many positive bands in line 5, with RG, in Figure 2 (c) and (d)? The previous westerns, (a) (b), in the same Figure, did not show all these recognize bands.

  • Sizes in the DNA ladder markers in Figure 2 (b,c,d) and also in the western bots (d,e) are missed. Please, add.

Author Response

Pleaase see attached file

Reviewer 3 Report

The authors of the manuscript „Adenovirus DNA polymerase loses fidelity on a stretch of eleven homocytidines” describe the construction of an HIV-1 vaccine based on an adenovirus vaccine vector, where they recognized the insertion of additional cytidines within a sequence of 11 homocytidines. They could reproduce this finding within an E. coli based setting. This paper indicates that long series of cytidines within an adenovirus based vaccine vector might lead to problems and should be avoided. Although this is an interesting finding, the data does not convincingly support the fact that this is only caused by the adenoviral DNA polymerase and not a problem of the clonal expansion of the virus. The manuscript contains major flaws and should be intensively revised, before any publication.

Major points:

- The focus of the paper and the title are not consistent. The main topic of the paper is the unsuccessful development of an HIV-vaccine candidate and not the loss of fidelity of the Adenovirus DNA polymerase. The authors should either adapt the title or shorten the description of the vaccine development to a minimum, as it is in this extent not necessary for the actual finding of the manuscript.

- As the detailed description of the unsuccessful vaccine development is not needed for the main finding of the manuscript, the data are not sufficient for an alone-standing publication. The main finding – the loss of fidelity of the adenovirus DNA polymerase – is not shown by the provided data and is therefore only an hypothesis. It could still be possible that the additional cytidine residues were already present in the BAC stock which might have been heterogenous. To prove that the insertion of cytidine residues into the vector sequence occurred during replication by the adenovirus polymerase, sequencing of the early (e.g. P1 and P2) passages would be necessary to prove a shift from 11 to 12-15 cytidines. Additionally, the comparison of a bacterial DNA dependent RNA polymerase to a viral DNA dependent DNA polymerase as a “reproduction” is highly speculative.

- The methods section lacks substantial information. Which cell line is used? The reader should be able to repeat the experiments based on the methods section, which is not possible in this case. Please describe the experiments in detail and indicate, e.g., used primer sequences for PCRs and the PCR conditions or refer to published data.

- The paper is written in an unscientific style, using, e.g., colloquial language. The intention of the authors and the needed information are meanwhile hard to understand. It should be revised by native speakers. Also, please define every abbreviation the first time it is used.

Minor points:

- Line 15: “inadvertently” is unscientific. Additionally, as described later in lines 310-311, one could have predicted the resulting cytidine sequence based on the sequence of the assembled conserved regions.

- Line 17: what is meant with “vaccine rescue”? This is also not scientifically written.

- Line 18: what are “blind passages”?

- Line 16-19: this sentence is very long and complicated and it is hard to get the information.

- Line 23: 1/25th of the virus stock is no indication of quantity. Please indicate the amount of used virus in, e.g., infectious viral particles.

- Line 23-24: “provide the quality signal to go ahead” is colloquial language and misleading

- Line 25-26: this finding was not demonstrated in the paper.

- Lines 31-38: References are missing

- Line 32: What are research-grade versions?

- Line 32: How are the vaccine candidates constructed?

- Line 39 and following: This section is rather methods than introduction

- Line 55: Which viral and cellular proteins?

- Line 55-56: Reference is missing

- Line 58: What are disabled adenovirus vectors? If “replication deficient” is meant, this correct designation should be used.

- Line 60: Which vaccine? The one from AstraZeneca is not the only COVID vaccine.

- Line 70: The methods are missing a chapter describing the used cell lines, their origin, and the cultivation conditions

- Lines 72-73: Please indicate or refer to the sequences and references of the used BAC and insert

- Line 73: the authors should provide exact amounts of reagents/DNA used and not approximately

- Line 75: How was the virus harvested?

- Line 79: The used cell line and its cell count are missing

- Line 81: “virus cultures” are with virus infected cell cultures?

- Line 82: For the methods used for quantification of virus concentration and for assessment of expression of transgene product and Western blot please refer to the corresponding methods section.

- Line 82: 24wp or 24 well-plate

- Lines 86-105: The PCR methods are missing the following substantial information: sequences of used primers, concentration of used primers, volume of used DNA sample, manufacturer of PCR MasterMix, conditions of the PCR reactions, explanation and methods description of how the product was verified (Agarose-Gel?)

- Line 108: How was lysis performed? Which lysis buffer was used? Which protease inhibitors? concentrations? how was the protein preparation performed?

- Lines 107-119: what was used as a loading control?

- Line 109: Please refer to the section where the procedure of the in vitro transcription/translation is explained

- Line 121: Please define which stages of amplification and clonal purification are meant? Which detergents were used for virus preparation?

- Line 122: Please describe how the harvesting, the detergent lysis and the depth filtration was performed or provide a reference

- Lines 121-127: the authors should describe in detail how they performed the TaqMan reporter-quencher system to calculate the virus particle concentration

- Lines 125-127: this is not a sentence

- Lines 122-124: The same information are missing as mentioned for the PCR (+ Sequence and Dye/Quencher of the probe). Additionally, it is important to describe in detail how the quantification was performed (Standard-Curves?). Is it really viral particles that were quantified or rather genomic equivalents or viral infectious particles?

- Line 126: No manufacturer is named, so the used protocol cannot be identified. Additionally, the sentence is incomplete.

- Line 128: What is the origin of the used mice? Animal approval number is missing.

- Line 130: What is the “or” following the “either”?

- Line 130-132: How were T cell responses analyzed?

- Line 136: Please refer to 2.10.

- Line 138: How was mass spectrometry performed? Which peptides were used?

- Line 137-140: Which peptides were used? The authors need to provide corresponding sequences and information why these particular peptides were employed

- Line 157-160: Which template DNA was used?

- Lines 162-168: Is rather methods than results. Additionally, please describe how the insertion was performed and how the sequence was confirmed. The authors should mention where the tHIVconsv6 ORF was inserted into the ChAdOx1 genome. Here a depiction of the BAC carrying the ChAdOx1.tHIVconsv6 would be highly appreciated.

- Lines 169-183: The authors should describe their results and not only the process

- Figure 2: loading control of (a) is missing; Why did the authors used different PCR-methods between the different passages to verify the transgene presence? The transgene presence checked in PCR is missing for (e)

- Figure 3: (c) The authors should indicate how many replicates were performed to calculate the standard deviations and which kind of error is displayed? (d) which primers were used for sequencing? only numbers were indicated but no sequences; (e) loading control is missing

- Line 170: What is meant with “virus rescue”?

- Line 178: Where are the PCR data shown?

- Line 184: In general, the figures are too small

- Line 190: infection of which cells?

- Line 185-196: This part is explained four times – in the methods section, in the second section of the results, in the figure legend and the figure itself – and is therefore redundant.

- Line 197-218: It is hard to understand which line of which figure is meant when, e.g., figure 2 c consists of three different figures with varying number of lines. Please give every figure a unique label. Additionally, it would be helpful to divide figure 2 in several figures, e.g., one for the agarose gels and one for the western blot results.

- Line 197: What are the additional bands Figure 2 a line 2 and 3 compared to 4?

- Line 205: Please indicate the origin of the research-grade vaccine.

- Line 223: How was the clinical batch material optimized?

- Line 225: “informed the engineering run” is colloquial language

- Line 226: What is ToxLot? Please describe the procedure in detail in the methods section.

- Line 226/figure 3b: The authors need to provide information on the AEX chromatogram. It is neither further explained in the results part, nor is there a methods section explaining how the data were acquired.

- Line 228: CD8+ T cells

- Line 231: colloquial language

- Line 238: The procedure of clone purifying and sequencing is not given in the methods section and no sequence of the used primers is given

- Line 239 and 240: this is only an hypothesis. The authors do not show data to prove that this is based on a loss in fidelity in the adenovirus polymerase; the conclusion is not reliable since it was not proofed if the AdV DNAP is responsible for the cytidine stretch elongation

- Line 241: Figure is again too small, no unique numeration of single figures, figure 3f is not mentioned in the text.

- Line 243: See comment of Line 18

- Line 245: What is WB?

- Line 256: Line 6 is the ladder, not the infected cells

- Line 268: Why was the fragment not directly sequenced in the VSS genome? The authors could have used Next generation sequencing techniques (e.g. Illumina Sequencing) to sequence their whole virus population. This would have also directly shown the different virus subpopulations.

- Line 269: 16 clones with 12 of what?

- Lines 268-272: I do not understand what the authors wanted to say

- Line 275: development

- Figure 5: Myc is underlined in red

- Line 277: This figure is hard to understand and not necessary for the manuscript.

- Line 286-296: the authors showed that also other polymerases has problems with its fidelity when there is a long stretch of the same nucleotide. So why it is purposed that the DNAP of the adenovirus is the problem and not the long cytidine stretch itself?

- Line 288: Please provide the sequences of the plasmids.

- Line 297: Numbering of the pictures is missing

- Line 309: See comment for Line 15

- Line 312: This is again only an hypothesis, see comments above.

- Line 360: The authors lack proof of their hypothesis, as described above.

-Line 366: “Lesson learned” is colloquial language

Round 2

Reviewer 1 Report

page 4, line 160: T25 should be corrected as T75